# No increase in use of hospitals for childbirth in Tanzania over 25 years: Accumulation of inequity among poor, rural, high parity women

Manuela Straneo [1,2,3]*, Lenka Beňová [3,4], Thomas van den Akker [2,5], Andrea B. Pembe [6], Tom Smekens [3], Claudia Hanson [1,4]

1 Department of Global Public Health, Karolinska Institutet, Solna, Sweden, 2 Athena Institute, VU University, Amsterdam, The Netherlands, 3 Institute of Tropical Medicine, Antwerp, Belgium, 4 London School of Hygiene &Tropical Medicine, London, United Kingdom, 5 Department of Obstetrics and Gynecology, Leiden University Medical Centre, Leiden, The Netherlands, 6 Department of Obstetrics and Gynaecology, Muhimbili University of Health and Allied Sciences, Dar es Salaam, United Republic of Tanzania

* manuela.straneo@ki.se

**Data Availability Statement:** The DHS dataset is freely available for use upon request at https://dhsprogram.com/data/available-datasets.cfm. After

## Abstract

Improving childbirth care in rural settings in sub-Saharan Africa is essential to attain the commitment expressed in the Sustainable Development Goals to leave no one behind. In Tanzania, the period between 1991 and 2016 was characterized by health system expansion prioritizing primary health care and a rise in rural facility births from 45% to 54%. Facilities however are not all the same, with advanced management of childbirth complications generally only available in hospitals and routine childbirth care in primary facilities. We hypothesized that inequity in the use of hospital-based childbirth may have increased over this period, and that it may have particularly affected high parity (≥5) women. We analysed records of 16,080 women from five Tanzanian Demographic and Health Surveys (1996, 1999, 2004, 2010, 2015/6), using location of the most recent birth as outcome (home, primary health care facility or hospital), wealth and parity as exposure variables and demographic and obstetric characteristics as potential confounders. A multinomial logistic regression model with wealth/parity interaction was run and post-estimation margins analysis produced percentages of births for various combinations of wealth and parity for each survey. We found no reduction in inequity in this 25-year period. Among poorest women, lowest use of hospital-based childbirth (around 10%) was at high parity, with no change over time. In women having their first baby, hospital use increased over time but with a widening pro-rich gap (poorest women predicted use increased from 36 to 52% and richest from 40 to 59%). We found that poor rural women of high parity were a vulnerable group requiring specifically targeted interventions to ensure they receive effective childbirth care. To leave no one behind, it is essential to look beyond the average coverage of facility births, as such a limited focus masks different patterns and time trends among marginalised groups.

the request has been approved, a de-identified dataset is made available. The authors confirm they had no special access or privileges that others would not have.

**Funding:** The authors received no specific funding for this work.

**Competing interests:** The authors have declared that no competing interests exist.

## Introduction

Maternal and neonatal mortality are unacceptably high in sub-Saharan Africa, with the poor disproportionately affected. The United Republic of Tanzania is no exception; in 2015, with a population of 51 million, there were an estimated 8,200 maternal [1] and 38,600 neonatal deaths, and 47,000 stillbirths [2]. Tanzania ranked amongst the ten countries worldwide with the greatest absolute numbers of maternal and newborn deaths and stillbirths. Under Sustainable Development Goal (SDG) 3 targets [3], global maternal mortality ratio (MMR) must be 70 per 100,000 live births in 2030, with no country above 140 per 100,000, which is far from the level of Tanzania in 2015 of 398 per 100,000 [1], while newborn mortality must reach 12 per 1,000, with a level of 22 per 1000 in Tanzania in 2015 [4]. Investments in childbirth care offer a triple return, with reductions in maternal and newborn deaths, and stillbirths.

Providing effective care in rural settings–where 68% of the Tanzania population resided in 2015—is key to addressing these deaths [4]. It is also essential to reduce mortality among the poorest, in line with the SDG commitment to leave no one behind [3]. Poverty remains more common in rural areas: in 2012, according to Household Budget Survey data, 33% of rural population lived below the national poverty line compared to 15% in urban areas [5]. Childbirth care coverage indicators such as facility births and skilled birth attendance rates are lower in rural compared to urban contexts [6]. A rural disadvantage on early neonatal mortality may be reversing in the country, due to the complex interaction of urban poverty and overburdened public health services primarily used by poorer women [7]. Nonetheless, the need to ensure effective services, including timely recognition and management of complications, in low-density, rural settings remains, with challenges linked to geographic accessibility, financial affordability and adequate facility capabilities including staffing [8]. High fertility in this context brings additional challenges to ensure services also serve the needs of high parity ($\geq 5$) women. Attention of health policy makers to the health-seeking and health needs of such women has been limited, despite their vulnerability due to being overburdened by household chores (many children at home, mostly agricultural work) [9], difficulties in accessing care due to rapidly progressing labour and being more prone to develop adverse outcomes, particularly postpartum haemorrhage [10].

Policies of newly independent Tanzania (1961) were shaped predominantly by the primary health care (PHC) philosophy (Fig 1 and S1 Table) [11]. Among the challenges at the time were a rapidly expanding, predominantly rural population (crude birth rate 5% in 1961) [12] and a health system heavily influenced by remnants of a colonial-era: a mostly urban, hospital-based, curative care structure. Regarding rural childbirth care, focus was on the expansion of availability of care. The vision of ensuring universal primary health care, elaborated in the Arusha Declaration in 1967 [13], was instrumental to reducing urban-rural inequities in availability of care. Provision of adequate and equitable childbirth care was among the aims of the first *Health Policy* [14], which specified its inclusion in PHC. Expansion of the PHC system (one dispensary per village, one health centre per ward) was detailed in the *Primary Health Care Development Plan* in 2007 [15]. Following disappointing outcomes with approaches focusing primarily on risk selection in the 1980s/1990s [9,16], and in line with strategies linked to the Millennium Development Goals, a shift in policy away from home births emerged after 2000 [11]. Since 2005, increasing facility birth coverage targets were set, aiming to increase skilled attendance at birth [17–19]. Another focus of policy has been the differentiation of care available at different levels of the health system. Despite the efforts to expand childbirth services and increase coverage of facility-based childbirth, there was some lack of clarity on the location (level of facility) where women were supposed to access childbirth care. Only the national antenatal card (Reproductive and Child Health Card, RCHC-4), which records

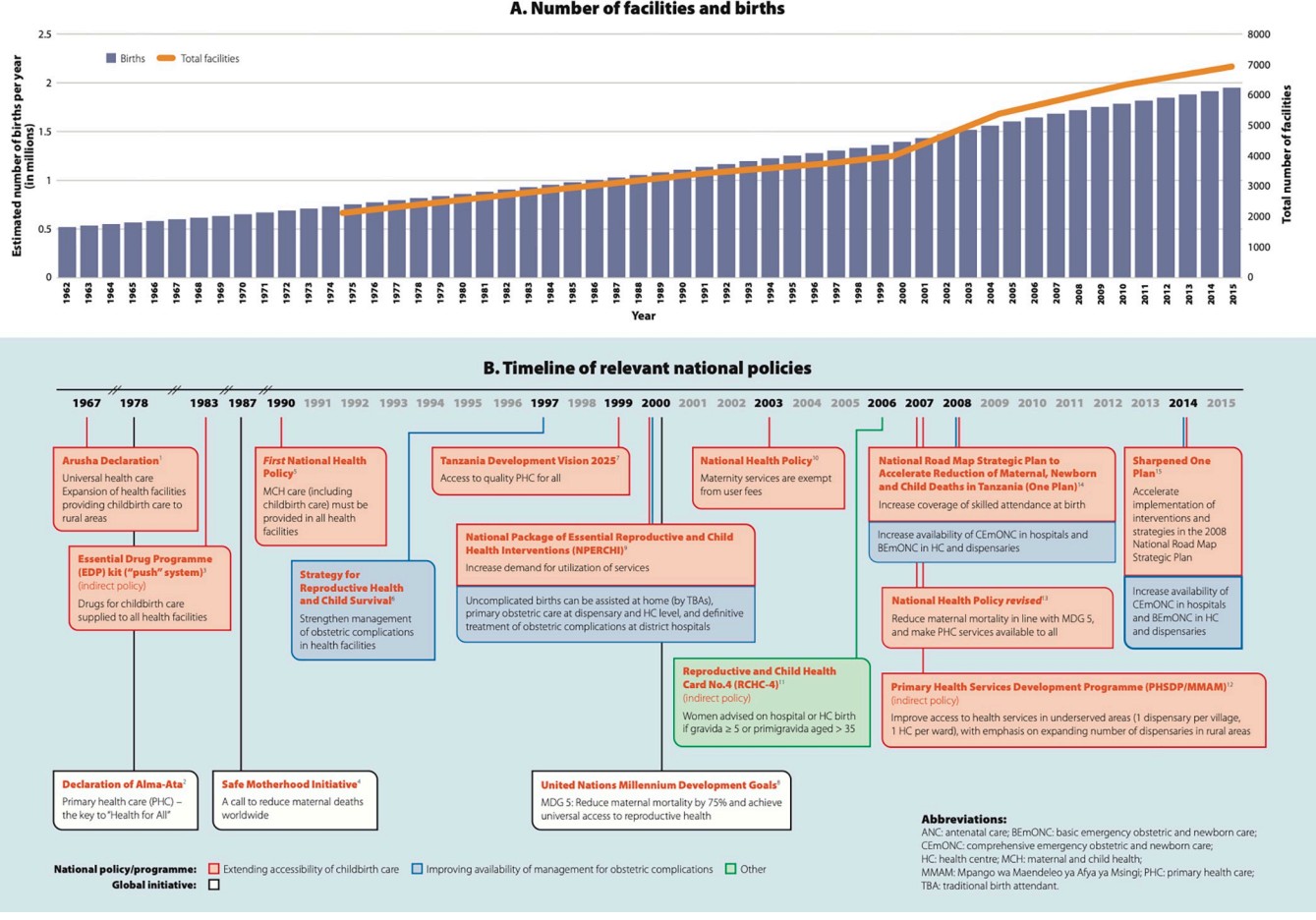

**Fig 1.** Trend of population and health facility numbers since independence in Tanzania (A) and timeline of relevant policies (B).

women's antenatal and birth care, identifies some higher risk groups who should give birth in higher level facilities, including women at parity ≥5 [20].

Over the 25-year period between 1991 and 2016, the health care system in the United Republic of Tanzania expanded in line with policy strategies (Table 1). The rise of PHC units (dispensaries +99%, health centres +162%) outpaced that of hospitals (+69%) during this time, though available figures include both rural and urban facilities, and comparisons are hampered by reclassification of rural areas as urban over time [6,21–25]. During this time, the population approximately doubled (from 26.1 to 51.5 million) [4] and births rose by 58% (Fig 1), corresponding to an average of 35,000 additional births per year.

**Table 1. Number of facilities and facilities/10,000 people over time, United Republic of Tanzania.**

| YEAR | HEALTH FACILITIES | | | | | | |
|------|------------|-------------------------|---------------|--------------------------|----------|-----------------------|-------|
|      | Dispensary | Dispensary /10,000 pop | Health Centre | Health Centre /10,000 pop | Hospital | Hospital /10,000 pop | Total |
| 1992 | 3000 | 11.13 | 273 | 1.01 | 152 | 0.56 | 3425 |
| 1996 | 3286 | 10.79 | 291 | 0.96 | 183 | 0.60 | 3760 |
| 1999 | 3500 | 10.71 | 302 | 0.92 | 195 | 0.60 | 3997 |
| 2004 | 4679 | 12.17 | 481 | 1.25 | 219 | 0.57 | 5379 |
| 2010 | 5469 | 12.33 | 633 | 1.43 | 240 | 0.54 | 6342 |
| 2015 | 5960 | 11.58 | 716 | 1.39 | 257 | 0.50 | 6933 |

Over the same 25-year period, rural women increasingly gave birth in facilities. Rural facility births rose from 45% of total births in 1992 [21] to 54% in 2015 [6]. However, obstetric capability varies at different levels of the health system [26], with advanced management of childbirth complications, such as caesarean sections and blood transfusion, generally only available at hospital level and PHC facilities providing routine childbirth care [8]. Challenges to quality of care in PHC facilities have been described repeatedly [26–28]. Lower use of higher capability, hospital-based childbirth among poor women may contribute to inequities in mortality.

This paper aims to assess how rural women's utilization of hospital-based childbirth care has changed over the period 1991–2016, during a time when substantial efforts were exerted to improve access through health system expansion. Based on the results of previous research [29], and in view of the link between poverty and high parity, we hypothesized an increase in inequity in hospital-based childbirth care use by wealth and variability by parity.

## Methods

### Data

We used Demographic and Health Surveys (DHS), nationally representative, household-based cross-sectional surveys, which provide socio-demographic and health utilisation data mainly for women and children. We included all Tanzania DHS surveys with a household wealth variable: 1996, 1999, 2004–5, 2010, 2015–16 [6,22–25]. The period analysed spans 1991–2016 (25 years), as DHS include data from women whose most recent birth was within 5 years of the survey. The 2004–5 survey is referred to as 2004 and the most recent survey is referred to as 2015, which correspond to the years when most data were collected.

### Study sample

From each dataset, we included records on the most recent live birth [in the five-year survey recall period) of women aged 15–49 at the time of survey living in rural areas in mainland Tanzania. Classification as 'rural' in DHS was based on census enumeration units [30].

### Outcome and independent variable definition

The outcome variable, 'self-reported location of the most recent live birth', included three categories: 'home', 'PHC facility' or 'hospital'. Home included 'respondent's home', 'other home' and 'en route to provider'. The second category included all PHC births (dispensaries, health centres, maternity homes, including DHS response "other facility"), whereas the third category included all hospital births (district, regional, referral or tertiary/university). Both the latter two categories (PHC and hospital) included public and private (non-profit/profit) facilities, as the DHS in Tanzania is one of few such surveys to disaggregate non-public facilities into hospitals and lower-level facilities. The five surveys had very little missingness in the birth location, <0.2% in all surveys except the earliest one (1996, 3.0%). Where the information on birth location was missing, it was re-coded as a PHC facility birth under the assumption that women's difficulty in reporting a facility name/level are more likely with lower-level facilities (many different types) rather than hospitals or home.

Independent/exposure variables of interest were household wealth as a proxy of maternal socio-economic status (SES) and parity. Household wealth quintiles are based on availability of durable household assets [31]. Due to under-representation of higher quintiles (richer & richest women] in rural areas [6], the two highest wealth quintiles were merged, to create four wealth groups (poorest, poorer, medium and richest). Parity refers to parity at index pregnancy, and was categorized as 0 (nullipara), 1–2, 3–4, and ≥5.

Other variables used in analysis include maternal age at index birth, maternal education and marital status, zone of residence, antenatal care (ANC) uptake during index pregnancy, and existence of obstetric risk factors [32]: multiple pregnancy, previous birth by CS, neonatal death in previous birth, previous baby died aged ≤12 months, preceding birth interval ≤12 months. Detailed description of the variables has been reported previously [29]. Zone of residence was included as a potential confounder to account for subnational variation. Classification of regions into zones is used by the Reproductive and Child Health section of the Ministry of Health. Zones have varied slightly over the period analysed; for this analysis, the eight zones described in the 2015–16 DHS were used [6].

As a measure of unmet obstetric need over time, the percentage of caesarean sections out of total births was examined in each survey. Women were classified as having had a caesarean section if they reported a facility birth and a section; the very few women who reported a caesarean section and a home birth were recoded as having had a vaginal birth.

## Statistical analysis

STATA IC 16 was used for analysis, using survey commands (svy) to account for complex design effect. Each survey dataset was analysed separately. Characteristics of each survey sample were analysed using percentages of outcome and exposure variables. Proportions of subgroups of women with exposure variables were studied for each level of outcome variable. Proportions of births at each level of outcome were calculated for combinations of wealth and parity.

Multinomial logistic regression was used for analysis, with place of birth as a categorical outcome (home, PHC or hospital), allowing us to include all births in a single model. The baseline outcome was birth in a PHC, thus the model produced odds ratios (ORs) of home versus PHC and hospital versus PHC births. Unadjusted and adjusted odds ratios for each level of the outcomes were estimated with 95% confidence intervals for each survey. Each model included, in addition to the variables of interest (wealth and parity), all *a priori* confounders (maternal age, education, zone of residence and ANC). Other variables were included if significant at p<0.05 in bivariate analysis.

As interaction between parity and wealth was of specific interest, a second logistic model was run for each survey, specifying an interaction term between the two variables. Adjusted ORs for each combination of wealth and parity were produced. Lastly, we obtained predicted percentages of women giving birth at each birth location for every survey by applying margins analysis to the logistic regression models with interaction.

## Ethical approval

The Demographic and Health Survey data collection procedures were reviewed and approved by the ICF Institutional Review Board [33]. The surveys are conducted by the Tanzanian government (National Bureau of Statistics or Ministry of Health, Community Development, Elderly, Gender and Children in 2015–16). Participants provided informed consent and assurance of confidentiality before the interviews [33]. Permission to study the datasets for this research was obtained in writing from the DHS program. The anonymised dataset was downloaded and used for this study.

# Results

## Characteristics of populations studied

We included live births of 16080 women (3285, 1176, 3709, 3454 and 4456 in 1996, 1999, 2005, 2010, 2015, respectively) living in rural areas of mainland Tanzania. Across the study period,

**Table 2. Main sample characteristics (birth location, wealth, parity) of rural women with a recent live birth from Tanzania in each survey analysed (DHS 1996, 1999, 2004, 2010 and 2015).**

| Survey year | 1996 (n = 3285) | | 1999 (n = 1176) | | 2004 (n = 3709) | | 2010 (n = 3454) | | 2015 (n = 4456) | |
|---|---|---|---|---|---|---|---|---|---|---|
| Variable/category | N | % | N | % | N | % | N | % | N | % |
| **Birth location** | | | | | | | | | | |
| Home | 1894 | 59.9 | 755 | 65.4 | 2172 | 58.8 | 1835 | 53.6 | 1815 | 41.3 |
| PHC | 659 | 19.3 | 200 | 15.4 | 876 | 23.6 | 996 | 28.8 | 1616 | 35.2 |
| Hospital | 732 | 20.8 | 221 | 19.2 | 661 | 17.6 | 623 | 17.6 | 1025 | 23.6 |
| **Wealth** | | | | | | | | | | |
| Poorest | 801 | 26.4 | 311 | 26.3 | 1051 | 26.8 | 923 | 25.0 | 1351 | 29.3 |
| Poorer | 757 | 22.7 | 279 | 25.5 | 982 | 25.8 | 988 | 28.3 | 1239 | 28.4 |
| Medium | 741 | 22.2 | 308 | 25.4 | 850 | 24.2 | 870 | 25.9 | 1104 | 25.2 |
| Richest | 986 | 28.8 | 278 | 22.9 | 826 | 23.2 | 673 | 20.8 | 762 | 17.1 |
| **Parity** | | | | | | | | | | |
| 0 | 663 | 20 | 224 | 19 | 689 | 18.6 | 556 | 16.4 | 963 | 22.2 |
| 1–2 | 1017 | 30.6 | 373 | 31.9 | 1302 | 34.8 | 1155 | 33.5 | 1342 | 30.5 |
| 3–4 | 743 | 22.7 | 275 | 23.6 | 843 | 22.6 | 873 | 25.3 | 1057 | 23.6 |
| ≥5 | 862 | 26.7 | 304 | 25.6 | 875 | 24.0 | 870 | 24.8 | 1094 | 23.6 |

home births reduced from 60% in 1996 to 41% in 2015. The corresponding 48% increase in facility births (40% to 59%), was due to a substantial increase of births in PHC (19% to 35%), with only a small overall rise in hospital births (21% to 24%, Table 2). All background characteristics (with 95% confidence intervals, CI) are reported in S2 Table. Women in the wealthiest group ranged between 29% in 1996 and 17% in 2015. Approximately one quarter of included women in each survey were parity 5 or higher (27% in 1996 and 24% in 2015).

While caesarean section percentages increased over this period (1.7%, 95% CI 1.3–2.3 in 1996 to 4.5%, 95% CI 3.8–5.4 in 2015) (S2 Table), they remained <5% of births in all years, indicating a still remaining unmet obstetric need in these rural populations.

In stratified analysis of birth location by wealth and parity (Table 3, with the full analysis in S3 Table), in all survey years examined, uptake of hospital-based childbirth increased with rising wealth and reduced with parity. Notably, among the women from wealthiest households, hospital childbirth rose from 31% to 45% in the period analysed, while it remained unchanged among women from the poorest households (13% in 1999, 16% in 2015).

## Logistic regression without wealth/parity interaction

**Home versus PHC birth.** In the adjusted analysis, we found that the variables of interest, wealth and parity, were significantly associated with the outcome (Table 4A). The odds of a home versus a PHC birth were lower in richest women compared to the baseline group (poorest), across all surveys (OR 0.47, 95% CI, 0.33–0.68 in 1996, and 0.34, 95% CI, 0.23–0.50 in 2015). Primiparous women in all survey years had lower odds of a home birth compared to the baseline parity 1–2 group, though in 2015 this did not reach statistical significance, while women at high parity had higher odds compared to the baseline in the three most recent surveys.

Regarding other variables in adjusted analysis, the odds of a home birth were greater in women with no education, or with no or <4 ANC visits in most survey years (S4 Table).

**Hospital versus PHC birth.** The adjusted odds of a hospital versus a PHC birth (Table 4B) were also significantly associated with wealth and parity. The odds were significantly higher in richest women compared to the poorest in the three most recent surveys, but

**Table 3. Bivariate analysis of outcome (home, primary care facility or hospital birth) by household wealth and parity among rural women with a birth in the five year period preceding the survey, by year of DHS survey.**

| Variable | 1996 (n = 3285) Home | PHC | Hospital | 1999 (n = 1176) Home | PHC | Hospital | 2004 (n = 3709) Home | PHC | Hospital | 2010 (n = 3454) Home | PHC | Hospital | 2015 (n = 4456) Home | PHC | Hospital |
|---|---|---|---|---|---|---|---|---|---|---|---|---|---|---|---|
| | % | % | % | % | % | % | % | % | % | % | % | % | % | % | % |
| | 95% CI | 95% CI | 95% CI | 95% CI | 95% CI | 95% CI | 95% CI | 95% CI | 95% CI | 95% CI | 95% CI | 95% CI | 95% CI | 95% CI | 95% CI |
| **Wealth** | | | | | | | | | | | | | | | |
| Poorest | 73.5 | 13.3 | 13.2 | 74.8 | 12.9 | 12.3 | 68.8 | 21.0 | 10.2 | 65.8 | 22.9 | 11.4 | 55.5 | 29.1 | 15.5 |
| | 67.4–78.8 | 10.3–16.9 | 9.9–17.5 | 67.2–81.1 | 8.5–19.2 | 7.6–19.1 | 64.6–72.7 | 17.9–24.5 | 8.0–12.9 | 61.6–69.7 | 19.2–26.9 | 9.1–14.2 | 50.5–60.3 | 25.4–33-0 | 12.3–19.1 |
| Poorer | 60.7 | 19.3 | 20 | 67.4 | 13.4 | 19.2 | 62.2 | 23.3 | 14.5 | 60.5 | 26.5 | 13 | 45.7 | 36.5 | 17.8 |
| | 55.0–66.2 | 15.6–23.6 | 16.2–24.4 | 57.3–76.0 | 9.6–18.5 | 12.0–29.4 | 57.7–66.5 | 19.6–27.5 | 11.8–17.8 | 55.7–65.2 | 10.6–15.8 | 10.6–15.8 | 41.5–50.0 | 32.7–40.5 | 14.9–21.1 |
| Medium | 61.1 | 21.5 | 17.5 | 66.5 | 16.7 | 16.8 | 59.6 | 22.8 | 17.6 | 49.5 | 33.0 | 17.5 | 35.5 | 39.7 | 24.9 |
| | 56.3–65.6 | 17.8–25.7 | 14.5–21.0 | 57.9–74.2 | 11.1–24.4 | 11.3–24.3 | 54.8–64.4 | 18.9–27.3 | 14.6–21.0 | 44.8–54.2 | 29.0–37.3 | 14.4–21.1 | 31.1–40.1 | 35.5–44.0 | 21.5–28.6 |
| Richest | 45.8 | 23.1 | 31.1 | 51.1 | 19.0 | 30.0 | 43.0 | 27.6 | 29.4 | 34.4 | 33.9 | 31.8 | 18.1 | 36.9 | 45 |
| | 41.1–50.7 | 19.3–27.4 | 26.6–36.0 | 39.6–62.5 | 13.1–26.6 | 20.1–42.1 | 38.1–48.0 | 22.9–32.8 | 24.9–34.4 | 30.1–38.9 | 29.8–38.3 | 28.0–35.8 | 14.1–23.0 | 31.9–42.2 | 39.4–50.7 |
| **Parity** | | | | | | | | | | | | | | | |
| 0 | 45.5 | 22.8 | 31.7 | 49.9 | 20.9 | 29.2 | 42.3 | 27.3 | 30.4 | 38.5 | 30.2 | 31.3 | 25.6 | 32.9 | 41.5 |
| | 40.3–50.9 | 19.0–27.0 | 0.27–36.9 | 40.1–59.7 | 14.1–29.9 | 21.9–37.8 | 37.6–47.2 | 23.0–32.1 | 26.3–34.8 | 33.5–43.8 | 26.1–34.7 | 27.0–35.9 | 22.2–29.4 | 29.2–36.8 | 37.1–46.1 |
| 1–2 | 60.4 | 17.7 | 21.9 | 63.5 | 15.8 | 20.7 | 57.9 | 24.7 | 17.4 | 51.0 | 31.3 | 17.7 | 40.6 | 37.3 | 22.1 |
| | 55.3–65.3 | 14.8–21.1 | 18.3–26.1 | 54.2–71.8 | 11.6–21.2 | 14.1–29.4 | 54.1–61.6 | 21.4–28.2 | 14.9–20.3 | 46.9-55-2 | 27.7–35.0 | 15.1–20.7 | 36.4–45.1 | 33.5–41.3 | 19.0–25.5 |
| 3–4 | 63.1 | 21.6 | 15.3 | 68.5 | 15.2 | 16.3 | 64.5 | 22.0 | 13.5 | 57.8 | 27.9 | 14.3 | 45.4 | 38.9 | 15.7 |
| | 57.9–68.0 | 17.7–26.1 | 12.1–19.1 | 61.0–75.2 | 10.7–21.1 | 10.9–23.7 | .60.1–68.7 | 18.7–25.7 | 10.8–16.9 | 53.2–62.3 | 24.3–32.0 | 11.7–17.3 | 41.1–50.0 | 35.0–42.9 | 12.9–18.9 |
| ≥5 | 67.3 | 16.5 | 16.2 | 76.4 | 11 | 12.7 | 67.9 | 20.5 | 11.7 | 62.6 | 25.4 | 12 | 52.7 | 30.9 | 16.4 |
| | 62.4–71.9 | 13.5–20.1 | 13.1–19.8 | 69.1–82.4 | 7.4–15.9 | 8.6–18.2 | 62.7–72.6 | 16.9–24.6 | 9.2–14.7 | 58.3–66.8 | 21.6–29.6 | 9.6–14.9 | 48.2–57.2 | 27.5–34.6 | 13.6–19.6 |

not in the earliest two. For this outcome, odds increased in primiparous women compared to the baseline (parity 1–2) over time: 1.79 (95% CI, 1.22–2.63, p = 0.003) in 2004 to 3.22 (95% CI, 2.34–4.43, p<0.001). In the first and last survey examined, high parity women had significantly lower odds, compared to women parity 1–2, of a hospital versus a PHC birth.

Among other variables studied, no education was associated with lower odds of a hospital birth in four surveys, while a previous caesarean section was associated with increased odds in three (S4 Table).

## Logistic regression with wealth/parity interaction

Our analysis using predicted percentages of women giving birth in each location for the extremes of parity and wealth by post-regression margins showed that hospital births increased over time in all wealth groups among nullipara (Table 5 and Fig 2; the complete analysis with 95% confidence intervals in S5 Table). Among the poorest, uptake of hospitals rose from 36% in 1996 to 52% in 2015, while among richest women, it increased from 40% to 59% in the same period. This contrasts with high parity women, where hospital births remained unchanged across all wealth levels. In poorest women, hospital births were 13% in 1996 and 12% in 2015, and among the wealthiest, they were 18% and 23% in the same years. The decline

**Table 4. Part a. Adjusted odds ratios of home versus primary health care birth in rural women with a live birth in the last five years, by year of DHS survey. Part b. Adjusted odds ratios of hospital versus primary health care birth in rural women with a live birth in the last five years, by year of DHS survey.**

| Variable | 1996 adjusted OR (95% CI)[1] | p-value | 1999 adjusted OR (95% CI)[2] | p-value | 2004 adjusted OR (95% CI)[3] | p-value | 2010 adjusted OR (95% CI)[4] | p-value | 2015 adjusted OR (95% CI)[5] | p-value |
|---|---|---|---|---|---|---|---|---|---|---|
| **Wealth** | | | | | | | | | | |
| Poorest | ref | | ref | | ref | | ref | | ref | |
| Poorer | **0.67 (0.46–0.98)** | **0.04** | 1.06 (0.63–1.78) | 0.83 | 0.9 (0.69–1.18) | 0.44 | 0.80 (0.62–1.03) | 0.08 | **0.75 (0.58–0.96)** | **0.02** |
| Medium | **0.61 (0.46–0.98)** | **0.003** | 0.85 (0.51–1.42) | 0.53 | 0.95 (0.73–1.23) | 0.68 | **0.55 (0.43–0.71)** | **<0.001** | **0.56 (0.43–0.74)** | **<0.001** |
| Richest | **0.47 (0.33–0.68)** | **<0.001** | **0.54 (0.31–0.96)** | **0.04** | **0.59 (0.42–0.82)** | **0.002** | **0.41 (0.30–0.56)** | **<0.001** | **0.34 (0.23–0.50)** | **<0.001** |
| **Parity** | | | | | | | | | | |
| 0 | **0.55 (0.38–0.98)** | **0.001** | **0.42 (0.23–0.79)** | **0.01** | **0.45 (0.31–0.65)** | **<0.001** | **0.71 (0.51–0.99)** | **0.05** | 0.84 (0.63–1.12) | 0.23 |
| 1–2 | ref | | ref | | ref | | ref | | ref | |
| 3–4 | 0.98 (0.38–1.78) | 0.87 | 1.3 (0.65–2.60) | 0.45 | **1.42 (1.11–1.80)** | **0.01** | **1.33 (1.03–1.73)** | **0.03** | 1.07 (0.82–1.39) | 0.64 |
| ≥5 | 1.27 (0.83–1.95) | 0.27 | 1.62 (0.62–4.34) | 0.33 | **1.71 (1.13–2.58)** | **0.01** | **1.72 (1.22–2.44)** | **0.002** | **1.54 (1.05–2.25)** | **0.03** |
| **Variable** | 1996 adjusted OR (95% CI)[1] | p-value | 1999 adjusted OR (95% CI)[2] | p-value | 2004 adjusted OR (95% CI)[3] | p-value | 2010 adjusted OR (95% CI)[4] | p-value | 2015 adjusted OR (95% CI)[5] | p-value |
| **Wealth** | | | | | | | | | | |
| Poorest | ref | | ref | | ref | | ref | | ref | |
| Poorer | 0.98 (0.66–1.44) | 0.91 | 1.49 (0.70–3.12) | 0.3 | 1.27 (0.88–1.84) | 0.20 | 0.98 (0.69–1.40) | 0.93 | 0.9 (0.66–1.23) | 0.44 |
| Medium | 0.74 (0.50–1.10) | 0.14 | 0.97 (0.39–2.38) | 0.91 | **1.51 (1.07–2.15)** | **0.02** | 0.96 (0.68–1.36) | 0.82 | 1.05 (0.78–1.41) | 0.84 |
| Richest | 1.11 (0.76–1.62) | 0.59 | 1.41 (0.61–3.27) | 0.42 | **1.77 (1.22–2.63)** | **0.004** | **1.51 (1.04–2.20)** | **<0.001** | **1.78 (1.26–2.50)** | **0.001** |
| **Parity** | | | | | | | | | | |
| 0 | 1.32 (0.91–1.93) | 0.15 | 0.86 (0.46–1.62) | 0.63 | **1.79 (1.22–2.63)** | **0.003** | **2.23 (1.53–3.24)** | **<0.001** | **3.22 (2.34–4.43)** | **<0.001** |
| 1–2 | ref | | ref | | ref | | ref | | ref | |

*(Continued)*

**Table 4.** (Continued)

| | 1996 | | 1999 | 2004 | | 2010 | | 2015 | |
|---|---|---|---|---|---|---|---|---|---|
| 3–4 | **0.47 (0.31–0.71)** | **<0.001** | 1.08 (0.47–2.50) | 0.75 (0.54–1.05) | 0.86 | 0.76 (0.54–1.06) | 0.1 | **0.46 (0.32–0.65)** | **<0.001** |
| ≥5 | **0.57 (0.33–0.96)** | **0.04** | 1.39 (0.52–3.76) | 0.74 (0.43–1.39) | 0.51 | 0.63 (0.38–1.05) | 0.27 | **0.59 (0.39–0.89)** | 0.10 ... 0.07 ... **0.013** |

[1] adjusted also for maternal age, maternal education, marital status, zone of residence, ANC, previous caesarean section.
[2] adjusted also for maternal age, maternal education, zone of residence, ANC.
[3] adjusted also for maternal age, maternal education, marital status, zone of residence, ANC, multiple live index birth, previous caesarean section.
[4] adjusted also for maternal age, maternal education, marital status, zone of residence, ANC, multiple live index birth, previous caesarean section.
[5] adjusted also for maternal age, maternal education, zone of residence, ANC, multiple live index birth, previous caesarean section.

**Table 5. Predicted margins (percentage) for each outcome level (home/PHC/hospital) birth in rural Tanzania by year of DHS survey and wealth in women at parity 0 and ≥5.**

| Outcome | Wealth | 1996 | | 1999 | | 2004 | | 2010 | | 2015 | |
|---|---|---|---|---|---|---|---|---|---|---|---|
| | | Parity 0 | Parity ≥5 | Parity 0 | Parity ≥5 | Parity 0 | Parity ≥5 | Parity 0 | Parity ≥5 | Parity 0 | Parity ≥5 |
| Home birth | Poorest | 44 | 75 | 57 | 71 | 40 | 76 | 41 | 71 | 28 | 58 |
| | Poorer | 46 | 70 | 56 | 80 | 36 | 73 | 42 | 66 | 28 | 56 |
| | Medium | 40 | 70 | 45 | 75 | 35 | 72 | 30 | 66 | 21 | 53 |
| | Richest | 36 | 62 | 44 | 59 | 24 | 62 | 24 | 53 | 16 | 30 |
| PHC birth | Poorest | 20 | 12 | 22 | 10 | 31 | 17 | 23 | 21 | 20 | 30 |
| | Poorer | 18 | 18 | 18 | 7 | 30 | 18 | 31 | 23 | 31 | 32 |
| | Medium | 25 | 18 | 35 | 12 | 28 | 17 | 30 | 25 | 28 | 35 |
| | Richest | 23 | 20 | 24 | 16 | 30 | 23 | 31 | 33 | 25 | 47 |
| Hospital birth | Poorest | 36 | 13 | 21 | 19 | 28 | 7 | 36 | 8 | 52 | 12 |
| | Poorer | 36 | 11 | 25 | 14 | 34 | 9 | 27 | 11 | 42 | 12 |
| | Medium | 35 | 12 | 20 | 13 | 37 | 10 | 40 | 8 | 51 | 12 |
| | Richest | 40 | 18 | 32 | 25 | 46 | 16 | 45 | 15 | 59 | 23 |

of hospital births as parity increases is evident in all surveys examined, is more pronounced among the poorest, and descending more gradually with parity in wealthier women.

Births in PHC facilities among nullipara remained constant across this period, while they increased over time in women with high parity. Use of PHC facilities more than doubled among wealthiest women (from 20% in 1996 to 47% in 2015). Home births declined in all wealth groups in the period examined. In nullipara, the decline among the richest took place earlier than among the poorest women. In high parity women, there was a greater reduction among richest women than among the poorest, and the greatest decline occurred between the two most recent surveys (Fig 2).

## Discussion

The study examined rural women's use of hospital-based childbirth in Tanzania over 25 years, a period characterized by health system expansion prioritizing PHC. We found large socio-economic disparities in hospital childbirth use between 1991 and 2016 against a background of rising facility births. While poorer women benefited from expansion of the PHC system for childbirth, richer women increasingly were able to use hospitals for childbirth. Poor rural women at first birth used hospital-based care less than the richest throughout the period

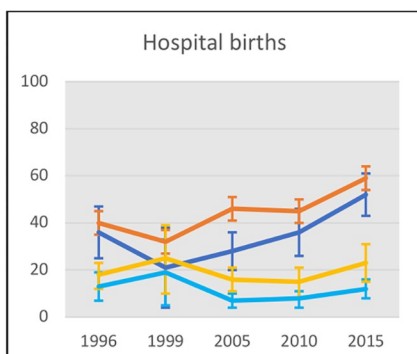
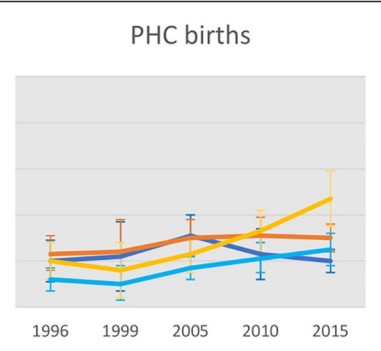
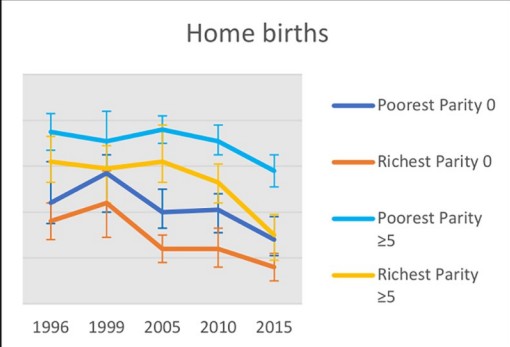

**Fig 2. Predicted percentages of births in hospital, PHC or at home, by year of survey, in poorest and richest women at parity 0 and parity ≥5.**

examined, but the highest equity-gap was seen in women at high parity ($\geq$5). Furthermore, this group had a very low hospital care use (around 10%), which has not increased over 25 years.

Our findings are of major public health significance: first, the use of hospitals for childbirth by high parity ($\geq$5) women in rural Tanzania remained very low in the period examined. There was no increase in uptake of hospital care despite increasing facility births in any wealth group, with a pro-rich gap in all survey years. Between 1991 and 2016 in a context with an unmet obstetric need, only one in ten women from the poorest group reported giving birth in a hospital. High parity women are at increased risk, mainly, though not solely, of postpartum haemorrhage [10,34], and this is of particular importance in rural contexts, where referral systems are fragile and lengthy [35–38]. Also, high parity was among recognized risk factors up to the 1990s [9], before the "all women are at risk" strategy was adopted because risk factors were poorly predictive of complications, and this may have remained in midwifery practice. We found no specific mention in policy documents about where high parity ($\geq$5) women should give birth, aside from the antenatal card [20] which indicates a health centre or hospital delivery for these women; this is not clear as some of the health centres do not provide surgical services and blood transfusion to curb complications when they occur. However, qualitative studies show that women receive pre-labour advice to give birth in a hospital [39]. Despite this, hospital uptake by this vulnerable group [9,40] has remained low. Possible explanations include individual and household factors, the dynamics of labour limiting women's capacity to travel far and altered risk perception [40].

Second, between 1991 and 2016, hospital-based childbirth use in rural Tanzania rose mainly among primigravidae. Those from richest households had greater uptake compared to the poorest in all survey years and the gap between the two extremes of wealth widened over time. Individual or household level factors have been described as important in explaining differences between women at first birth and successive ones [27,41], but broader country or health system factors may come into play [42]. Though we did not identify a clear indication in policy documents in relation to where women should give birth to their first child, with only the antenatal card specifying referral to a hospital or health centre for primigravidae aged 35 years or more [20], it is likely that women received advice to give birth in hospital [43]. This is consistent with previous studies, which found that nulliparous women were more frequently represented at hospital level [44] and were more likely to bypass PHC facilities [43].

Third, our findings show that solely looking at the increase in the percentage of all births occurring in any health facility (corresponding to decline in home births) masks important patterns and trends by parity and socio-economic background. Such differentials must be considered in health care organisation aiming to provide for all, particularly in rural areas. Facility births have increased globally, following strategies to increase skilled attendance at birth [45]. In some countries, as part of the strategy to improve coverage of facility births, there were policies restricting home births [42,46]. In Tanzania there are reports of homebirth fines [47] applied at local level (village or district), but to the best of our knowledge there were no national policies restricting home births. However, traditional birth attendants' care of childbirth at home was included in policy only up to the year 2000 [11].

Our study proposes that in rural Tanzania, the increase in facility births was driven by childbirth in hospitals among nulliparous women and in PHC facilities among women at parity $\geq$5. There is evidence of lower quality of maternity care in PHC facilities in rural Tanzania [26,43,48]. Across low-income countries, a two-tiered system for childbirth has been described [49]. Identifying vulnerable groups that do not reach the recommended level of care is a necessary step to propose strategies to enable them to obtain effective care.

Health systems need to be designed in a way that they contribute to equity in health, by specifically addressing the needs of disadvantaged groups [50,51]. To address the needs of poor women, particularly at high parity, we outline three recommendations.

First, adoption of strategies to close the equity gap in hospital uptake: evidence is scant on what is effective [52]. In India, the combination of strengthened PHC obstetric care and conditional cash transfers contributed to reduce inequity in institutional births [53]. Other possible interventions may include vouchers [54,55], including transport vouchers, and promotion of maternity waiting homes [56]. Improving uptake by poorest women at high parity needs to address the additional layer of who cares for the children at home, which may require community support.

Second, to account for the diverse needs of women [57], the configuration of rural childbirth care needs increased attention within the framework of universal health coverage. Poor women at high parity constitute a "pocket of vulnerability" [57], whose voice is not often heard [9]. It can be argued they have voted with their feet, uptaking services in PHC facilities, despite antenatal hospital referral [35]. Rural SSA is an arduous context for childbirth care, as a balance must be found between population access and the capacity of facilities to provide effective childbirth care [58]. Centralization of childbirth care to hospitals has been proposed to reduce disparities in quality of maternity care [37,59], similarly to high income countries. However, in this rural context, one quarter of women were at parity ≥5 in all survey years studied. In view of the combination of a substantial proportion of high parity women and low rural population density, centralization of childbirth care to hospitals -as proposed by Kruk et al [37]- may be accompanied by a rise in home births among poorer women if no strong supporting measures are concurrently implemented. Strengthening maternal care in selected PHC facilities, with a concurrent, stepwise shift away from low-volume units guided by geospatial data [42,60], would allow close monitoring of potential negative effects of increased home births. Though some degree of centralization may be feasible without compromising accessibility [61], solutions should be adapted to the context and based on evidence [16,62]. Strengthening maternal health care in health centres to provide safe caesarean section and blood transfusion as part of emergency obstetric care is one strategy pursued by the Tanzanian government [63], though its effectiveness on mortality reduction has not yet been confirmed. The importance of emergency transport cannot be overlooked, taking account of day/night and seasonal variations in roads' network. A participatory research approach for the complex problem of childbirth care for rural, poor, high parity women may bring forward novel solutions and mitigate challenges hindering use of facility childbirth services.

Lastly, the facility births' indicator groups different levels of maternal care, which should be analysed separately to best identify trends in health system uptake.

## Strengths and limitations

The main strength of this study is that it sheds light on inequities within the health system for childbirth, identifying poor high parity (≥5) women as a disadvantaged group in up-taking hospital-based childbirth over 25 years. The novelty of the research question, whether inequity depends on parity, is an additional strength. It is a comprehensive study on uptake of childbirth care across twenty-five years, beyond the facility births indicator, informed by nationally representative data. Studying the health system in Tanzania, which has strong emphasis on PHC, allows to draw conclusions that may contribute to policies in other countries rolling out rural childbirth services. Additionally, details collected in the Tanzanian DHS on type of birth facility both in public and non-public system (the latter is unavailable in most other SSA countries' surveys), provide a more precise picture avoiding misclassification between hospitals and PHC facilities.

Key limitations are derived from methodological characteristics of DHS data, such as reporting bias, and have been described in detail [46]. Subdivision of facilities into PHC and hospitals, which reflects the national classification [43], does not allow to capture variations in obstetric capability at the two levels of care. Grouping together health centres and dispensaries did not allow for differentiation of higher level of care at health centre level. Though there is a national programme to upgrade health centres to provide comprehensive emergency care [63], the facilities providing the full range of functions were very limited in the period studied. Analysis of most recent births only may have improved accuracy of recall, but it also may have resulted in undersampling of births in the recall period. As poor women tend to have high parity, the study may have undersampled births of poor women.

## Conclusion

In rural Tanzania, over a period of 25 years characterized by health system expansion prioritizing PHC, there was no closing of the equity gap in hospital-based childbirth. Use of hospitals by poor women at high parity remained very low, around 10%.

The configuration of rural childbirth care, including any centralization, should in particular address the vulnerability of the poor at high parity. Interventions should be targeted to specific settings and should also improve linkages between the levels of care.

## Supporting information

**S1 Table. Key policies in Tanzania which influenced where women can give birth from independence (1961) to 2015.**
(XLSX)

**S2 Table. Main sample characteristics of rural women with a recent live birth from Tanzania in each DHS survey analysed.**
(XLSX)

**S3 Table. Bivariate analysis of outcome (home, primary care facility or hospital birth) by independent variables among rural women with a recent live birth, by year of DHS survey.**
(XLSX)

**S4 Table. Crude and adjusted odds ratios of home versus PHC birth (left) and hospital versus PHC birth (right) in each year of DHS survey, in rural women with a recent live birth.**
(XLSX)

**S5 Table. Predicted margins (percentage) with 95% CI for each outcome level (home/PHC/hospital) birth in rural Tanzania by year of DHS survey and wealth group in women at parity 0 and ≥5.**
(XLSX)

## Acknowledgments

We wish to thank the women who agreed to be interviewed during the surveys, the interviewers, and the DHS program. The latter kindly provided access to the surveys' databases for the present analysis.

We acknowledge the support of Green Ink Publishing Services Ltd in the preparation of Fig 1.

## Author Contributions

**Conceptualization:** Manuela Straneo, Lenka Beňová, Thomas van den Akker, Claudia Hanson.

**Formal analysis:** Manuela Straneo.

**Methodology:** Manuela Straneo, Lenka Beňová, Tom Smekens, Claudia Hanson.

**Supervision:** Lenka Beňová, Thomas van den Akker, Andrea B. Pembe, Claudia Hanson.

**Writing – original draft:** Manuela Straneo.

**Writing – review & editing:** Manuela Straneo, Lenka Beňová, Thomas van den Akker, Andrea B. Pembe, Tom Smekens, Claudia Hanson.

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
