## [Decision Letter · Decision Letter 0]

14 Jun 2022

PGPH-D-22-00216

No increase in use of hospitals for childbirth in Tanzania over 25 years: accumulation of inequity among poor, rural, high parity women

Dear Dr. Straneo,

Thank you for submitting your manuscript to PLOS Global Public Health. After careful consideration, we feel that it has merit but does not fully meet PLOS Global Public Health’s publication criteria as it currently stands. Therefore, we invite you to submit a revised version of the manuscript that addresses the points raised during the review process.

My compliments on a well written manuscript. One of our reviewers had no suggested edits.

Please see below for the edits suggested by the other reviewer -

"Under methods and lines 131-135 need revising because line 133 refers to "the third and last survey which were referred to as 2004 and 2015..." This could be rephrased because it is not clear what the word "last" refers to.

Under ethical approval line 197, there seems a need to specify which ethics committees, boards or official titles granted permissions for the study. Ethics approval by "DHS program", line 200, appears too broad in this context and may be revised.

Results, line 202, are derived from already collected data from surveys covering the reported 25 years as indicated. It is assumed that the data was raw in that it was never analysed and interpreted to address the current research question. in that sense, this is viewed as primary scientific research.

Under line 382 strengths and limitations, inequity and the vulnerable population at risk was addressed adequately. Nevertheless, the preference for the current statistical tool may need to comment on as SPSS appears superior in social and medical sciences."

I recommend that you address the first three issues that were brought up by the reviewer. I understand the thoughts behind the last recommendation, please do not feel obliged to address it.

Please submit your revised manuscript by . If you will need more time than this to complete your revisions, please reply to this message or contact the journal office at globalpubhealth@plos.org. Please include the following items when submitting your revised manuscript:

We look forward to receiving your revised manuscript.

Kind regards,

Shailendra Prasad, MD, MPH

Academic Editor

Journal Requirements:

1. Please amend your detailed Financial Disclosure statement. This is published with the article, therefore should be completed in full sentences and contain the exact wording you wish to be published.

2. Please update the 'Competing Interests' statement with this "The authors have declared that no competing interests exist".

3. In the online submission form, you indicated that “Secondary analysis of publicly available data”. All PLOS journals now require all data underlying the findings described in their manuscript to be freely available to other researchers, either 1. In a public repository, 2. Within the manuscript itself, or 3. Uploaded as supplementary information.

**Comments to the Author**

1. Does this manuscript meet PLOS Global Public Health’s publication criteria? Is the manuscript technically sound, and do the data support the conclusions? The manuscript must describe methodologically and ethically rigorous research with conclusions that are appropriately drawn based on the data presented.

Reviewer #1: Yes

Reviewer #2: Yes

2. Has the statistical analysis been performed appropriately and rigorously?

Reviewer #1: Yes

Reviewer #2: Yes

3. Have the authors made all data underlying the findings in their manuscript fully available (please refer to the Data Availability Statement at the start of the manuscript PDF file)?

Reviewer #1: Yes

Reviewer #2: Yes

4. Is the manuscript presented in an intelligible fashion and written in standard English?

Reviewer #1: Yes

Reviewer #2: Yes

5. Review Comments to the Author

Reviewer #1: Under methods and lines 131-135 need revising because line 133 refers to "the third and last survey which were referred to as 2004 and 2015..." This could be rephrased because it is not clear what the word "last" refers to.

Under ethical approval line 197, there seems a need to specify which ethics committees, boards or official titles granted permissions for the study. Ethics approval by "DHS program", line 200, appears too broad in this context and may be revised.

Results, line 202, are derived from already collected data from surveys covering the reported 25 years as indicated. It is assumed that the data was raw in that it was never analysed and interpreted to address the current research question. in that sense, this is viewed as primary scientific research.

Under line 382 strengths and limitations, inequity and the vulnerable population at risk was addressed adequately. Nevertheless, the preference for the current statistical tool may need to comment on as SPSS appears superior in social and medical sciences.

Reviewer #2: Study is well structured with appropriate statistic analysis. Study question is clear and important to public health polices changes and may be used in practice. Findings seems quite counterintuitive though most valuable. Long observation period and focus on most vulnerable people are strong points of the publication.

6. PLOS authors have the option to publish the peer review history of their article (what does this mean?). If published, this will include your full peer review and any attached files.

**Do you want your identity to be public for this peer review?** For information about this choice, including consent withdrawal, please see our Privacy Policy.

Reviewer #1: **Yes: **John Mukuka Musonda

Reviewer #2: No

---

## [Editor Report · Decision Letter 1]

18 Jul 2022

No increase in use of hospitals for childbirth in Tanzania over 25 years: accumulation of inequity among poor, rural, high parity women

PGPH-D-22-00216R1

Dear Dr Straneo,

We are pleased to inform you that your manuscript 'No increase in use of hospitals for childbirth in Tanzania over 25 years: accumulation of inequity among poor, rural, high parity women' has been provisionally accepted for publication in PLOS Global Public Health.

Best regards,

Shailendra Prasad, MD, MPH

Academic Editor

Thank you for addressing the concerns. This revised manuscript reads better and I have recommended that this be accepted to publish.

Congratulations!